# Quercetin alleviates ulcerative colitis via regulating gut microbiota and tryptophan metabolism

Man Xiong,[1] Wenli Kuang,[2] Zhenyang Liu,[2] Ruoyi Tong,[1] Xu Deng,[1] Nan Wang,[3] Xingzhi Wan,[4] Mengyuan Feng,[4] Yan Luo,[1] Bin Zhang,[5] Zhaofu Zhang,[2] Fanghao Zheng[6]

**ABSTRACT**  Quercetin, a natural flavonoid in traditional Chinese medicinal plants, has shown promise in alleviating ulcerative colitis symptoms despite uncertainties about its exact mode of action. This study explored how quercetin influences tryptophan breakdown and gut bacterial populations in mice with chemically induced colitis. The treatment demonstrated measurable improvements—normalizing body weight, reducing spleen enlargement, lowering clinical severity scores, preserving colon structure, and healing tissue damage. Through advanced microbiome profiling and metabolic analysis, researchers observed increased populations of helpful gut microbes alongside higher concentrations of tryptophan byproducts. These biochemical shifts stimulated the aryl hydrocarbon receptor system, which plays a key role in restoring gut lining integrity. The collective evidence points to quercetin's therapeutic potential through its dual action on microbial ecology and tryptophan-derived signaling pathways.

**IMPORTANCE**  Ulcerative colitis is a chronic inflammatory disease with limited effective therapeutic options. In this study, quercetin—a flavonoid commonly found in traditional Chinese medicinal herbs—was shown to relieve colitis symptoms by reshaping gut microbiota and restoring tryptophan metabolism. Notably, the increase in indolelactic acid, a key microbial metabolite, led to activation of the aryl hydrocarbon receptor, which supports intestinal barrier integrity and dampens inflammation. These findings reveal a gut microbiota-derived metabolite-host signaling axis as a central mechanism of action, highlighting the potential of quercetin as a microbiota-targeted therapeutic approach for UC.

**KEYWORDS**  ulcerative colitis, quercetin, microbiota disorders, tryptophan metabolism, aryl hydrocarbon receptor

Ulcerative colitis (UC) is a persistent inflammatory condition affecting the bowel, primarily targeting the rectum and sigmoid colon with diffuse mucosal inflammation. Patients typically experience bloody, mucus-filled stools, frequent diarrhea, unexplained weight loss, and chronic abdominal pain (1). The worldwide incidence of UC has risen dramatically in recent years, affecting approximately 5 million people globally as of 2023 (2). While current treatment options include 5-aminosalicylic acid (5-ASA) compounds, corticosteroids, immunosuppressive drugs, and biologics (3), finding a safe and highly effective medication continues to be a pressing medical challenge.

Emerging studies have established a clear link between UC and alterations in gut microbiota composition (4, 5). Notably, microbial imbalances can worsen UC symptoms (6), suggesting that modulating intestinal flora could represent an effective treatment approach (7, 8). Beyond the bacterial populations themselves, microbial products also play a significant role in the development of inflammatory bowel disease (IBD).

**Peer Reviewers** Zibin Lu, School of Traditional Chinese Medicine in Southern Medical University, Guangzhou, China; Qinhai Ma, Guangzhou Medical University, Guangzhou, China

Address correspondence to Fanghao Zheng, zhengfh@fshtcm.com.cn, Bin Zhang, zhangbin0993@qdu.edu.cn, or Zhaofu Zhang, zhangzhf36@mail2.sysu.edu.cn.

Man Xiong and Wenli Kuang contributed equally to this article. The author order was determined based on their contribution to the article.

The authors declare no conflict of interest.

Tryptophan-derived metabolites, such as indole, indolelactic acid (ILA), and indole-3-carbinol, act as endogenous ligands of the aryl hydrocarbon receptor (AhR). Binding to the AhR activates the receptor, which is expressed in intestinal epithelial cells and recognized as essential for maintaining the integrity of the intestinal mucosal barrier (9, 10). The AhR has widespread expression in intestinal epithelial cells. It is acknowledged as an essential element for upholding the intactness of the intestinal mucosal barrier (11). Nevertheless, research examining the interplay between microbiota, tryptophan metabolism, and AhR activation in the context of UC remains relatively sparse and necessitates further comprehensive investigation.

Quercetin, chemically named 3, 5, 7, 3′, 4′-pentahydroxyflavone and often abbreviated as Qu, belongs to the class of natural flavonoids. It is renowned for possessing multiple beneficial characteristics such as anti-inflammatory, anti-ulcer, anti-tumor, and antioxidant activities (12). Notably, quercetin has demonstrated significant anti-inflammatory effects, which contribute to its potential benefits in enhancing systemic metabolic function. For instance, it has been reported to mitigate interstitial inflammation and deformities associated with arthritis in the context of diabetic nephropathy (13–15). The literature also indicates that quercetin regulates the alternative activation of macrophages by modulating the balance of STAT1/PPARγ, thereby exerting a mitigating effect on colitis (16). Nevertheless, studies regarding the impact of quercetin on microbiota and microbial metabolites within the scope of UC are still scarce. Consequently, a more thorough and detailed investigation is warranted. This study aimed to examine how quercetin might mitigate dextran sulfate sodium (DSS)-induced colitis in mice by modulating gut microbiota and influencing tryptophan metabolism.

## MATERIALS AND METHODS

### Mice

The male C57BL/6 mice, aged between 6 and 8 weeks and weighing in at 20 to 22 g, were sourced from BesTest Bio-Tech Co., Ltd. in Zhuhai, China. They were kept in a meticulously controlled, SPF barrier facility where the temperature was kept at a steady 22°C±1°C. The mice were subjected to a consistent 12-h light/dark schedule and had free reign over their food. The Institutional Animal Care and Use Committee (IACUC) of Huateng Biomedical Science and Technology Co., Ltd. in Shenzhen, China, had greenlit all experimental procedures involving these animals. The go-ahead was given under protocol number B202403-4.

### Animal experiments

To investigate how quercetin affects colitis, we induced the condition in mice using a 3% DSS solution administered continuously for a week. We randomly divided 30 C57BL/6 mice into five groups of six. These groups were (i) a normal control group (NC); (ii) a colitis model group induced by DSS; (iii) a group treated with 5-aminosalicylic acid at 200 mg/kg; (iv) a low-dose quercetin group (L-Qu) receiving 50 mg/kg; and (v) a high-dose quercetin group (H-Qu) getting 100 mg/kg.

To investigate how gut bacteria might play a part in managing UC, researchers split the mice into four distinct groups: a control group (NC) given regular water, a group that received fecal microbiota transplants from untreated mice (DSS-FMT), a group that received fecal microbiota transplants from mice treated with quercetin (Qu-FMT), and a group treated with quercetin on its own (Qu, at 100 mg/kg). The latter three groups, however, were given drinking water containing 3% DSS for a solid week to induce colitis. On top of this, the DSS-FMT group was given fecal suspension from DSS donor mice via gavage, while the Qu-FMT group got fecal suspension from Qu donor mice the same way. To establish a germ-free mouse model, the mice were given a cocktail of antibiotics (Abx)–vancomycin hydrochloride (0.5 g/L), ampicillin sodium (1 g/L), metronidazole (1 g/L), and neomycin sulfate (1 g/L)– via gavage for 7 days straight.

To assess the role of tryptophan metabolites in quercetin's therapeutic effects on UC, the mice were divided into three randomized cohorts: a NC group, a DSS model group, and an ILA intervention group. The latter received ILA supplementation at 50 mg/kg to evaluate its potential benefits.

To further investigate AhR's involvement in quercetin's therapeutic effects on UC, we performed additional animal studies. A cohort of thirty mice was randomly assigned to five experimental groups: a NC group, a DSS-induced colitis model group, a group receiving only the AhR inhibitor CH223191 (10 mg/kg via intraperitoneal injection), a combination therapy group receiving both quercetin and CH223191 (10 mg/kg each, administered intraperitoneally), and a quercetin-only treatment group (Qu).

## Disease activity index (DAI)

Mice were observed for changes in body weight, the texture of their feces, and any bleeding around the anus on a daily basis. These observations were then pooled together to calculate the Disease Activity Index (DAI) scores using the following criteria: (i) weight loss (0 points for no loss, one point for a 1%–5% drop, 2 points for 5%–10% loss, 3 points for 10%–20% reduction, and four points for a 20% decrease); (ii) diarrhea severity (0 points for normal, two points for slightly loose stools, and four points for watery diarrhea); (iii) blood in the stool (0 points for no bleeding, two points for minor bleeding, and four points for significant bleeding).

## Hematoxylin-eosin (H&E) staining

To ensure consistency, all mouse liver tissue samples were carefully harvested from identical anatomical sites before being preserved in 4% paraformaldehyde for a full day. The specimens underwent dehydration and clearing processes, followed by wax infiltration and paraffin embedding. These embedded blocks were then cured in a 65°C oven for 4 h. Post-curing, the paraffin was removed by melting, and the sections were cleared in xylene before being gradually rehydrated through an ethanol gradient. The prepared tissue slices were then stained with hematoxylin and eosin for histological analysis. After staining, the slides were mounted with neutral gum and stored under dry, ambient conditions. For evaluation, multiple random fields of view were selected, and images were acquired at room temperature.

## Immunohistochemistry

Following deparaffinization, the tissue sections underwent a series of ethanol washes, with three 5-min immersions in absolute ethanol. Next, the samples were transferred to a pH 6.0 citrate buffer solution and subjected to 15 min of heat-induced epitope retrieval in a boiling water bath. To quench endogenous peroxidase activity, the sections were treated with a blocking agent for 25 min at ambient temperature. Primary antibody incubation proceeded overnight at 4°C, after which matched secondary antibodies were applied for a 50-min room temperature reaction. Chromogenic development was achieved using DAB substrate for 15 min at room temperature. The sections were then counterstained with hematoxylin for 3 min, followed by sequential dehydration through an ethanol gradient and xylene clearing. Mounting with neutral balsam completed the preparation process prior to microscopic imaging.

## Western blotting

The proteins in the mouse colonic tissue were extracted using RIPA lysis buffer (Beyotime, product number P0013B), and the total protein content was determined by the BCA method (Beyotime, product number P0010). Protein samples of equal mass were separated by SDS-PAGE electrophoresis and transferred to a PVDF membrane (0.22 µm, Millipore, Germany). After being blocked at room temperature for 1.5 h, four primary antibodies were added successively and incubated at 4°C for 16 h: the tight junction protein Occludin (1:1,000, Proteintech, 66378-1-Ig), Claudin-1 (from the same brand,

1:1,000), Cytochrome P450 1B1 (Proteintech, 18505-1-AP, 1:1,000), and the internal reference GAPDH (Affinity, 13050-1-AP/T0004, 1:10,000). Finally, a highly sensitive ECL chemiluminescent substrate (Beyotime) was used for the color reaction. The signal was captured by a Tanon gel imaging system (Shanghai), and the detection was quantified using ImageJ image analysis software.

## Quantitative real-time polymerase chain reaction (qRT-PCR)

In this study, we employed the Trizol method to pull out all the RNA from our tissue samples. Once we had that RNA in hand, we used a fancy, ultra-micro spectrophotometer to check its quality (specifically, the A260/A280 ratio) and figure out how concentrated it was. Next, we used reverse transcriptase to turn that RNA into its cDNA counterpart. For gene amplification, we followed the kit instructions to the letter. Our PCR reaction was performed as follows:: a quick 30-s pre-denaturation step at 95°C, then 40 cycles of amplification. Each cycle involved a 5-s blast at 95°C for high-temperature denaturation, followed by 30 s at 60°C for annealing and extension. To keep our data on the up-and-up, we went with GAPDH as our trusty endogenous control. You can find all the details about the specific primer sequences we used in Table 1.

## Fecal metagenome sequencing

DNA concentration was measured with a Qubit 2.0 fluorometer, following the Qubit dsDNA Assay Kit's protocol to the letter. Next, each DNA sample, with its concentration adjusted to 1 µg, was randomly fragmented to roughly 350 bp using an ultrasonic disruptor. Following fragmentation, we went through a series of steps to prep the samples for sequencing: think end-repair, adding an A-tail, and attaching sequencing adaptors. After cleaning everything up, we amplified the fragments using PCR. And with that, the sequencing library was good to go. To keep track of things, we tagged each sample with a unique index. The PCR products were then purified using the AMPure XP system. To check the size distribution of the library, we ran it on an Agilent 2100 Bioanalyzer. After that, we used real-time PCR to get a handle on the quantity of DNA. Finally, it was time to sequence the library.

## Analysis of tryptophan metabolites

A 200-µL aliquot of plasma was prepared, to which 10 µL of an internal standard solution (4 µg/mL) was added. Next, 400 µL of ice-cold methanol with 0.1% formic acid was introduced. The combined solution was vortexed thoroughly for 30 s before undergoing centrifugation at 13,000 rpm for 10 min at 4°C. After centrifugation, the clear supernatant was carefully pipetted and dried down using nitrogen gas. The dried residue was then redissolved in 200 µL of a 0.1% formic acid aqueous solution and passed through a hydrophilic membrane filter. These prepared samples were now suitable for further analytical procedures.

## Statistical analysis

Data analysis was carried out using GraphPad Prism (version 9.5.0), and we have reported our findings as the mean ± standard error of the mean (mean ± SEM). To see how the groups stacked up against each other, we ran either an ANOVA or a Kruskal-Wallis test, depending on what the data looked like. Anything with a $P$-value less than 0.05 was considered a statistically significant difference. For the trans-omics stuff, looking at both

**TABLE 1** Primer of the study

| Gene | Forward primer sequence (5′–3′) | Reverse primer sequence (5′–3′) |
|---|---|---|
| IL-1β | TGCCACCTTTTGACAGTGATG | TGATGTGCTGCTGCGAGATT |
| IL-6 | GGGACTGATGCTGGTGACAA | ACAGGTCTGTTGGGAGTGGT |
| TNF-α | TAGCCCACGTCGTAGCAAAC | TGTCTTTGAGATCCATGCCGT |

gut microbiota and metabolomes, we mostly relied on Spearman's rank correlation. Plus, we dove into the nitty-gritty of microbiota-metabolite relationships using the "psych" and "pheatmap" packages in R (version 4.3.1).

## RESULTS

### Quercetin improved the symptoms of UC in DSS-induced mice

Acute colitis in mice was evoked by orally administering 3% DSS continuously for seven days, followed by daily oral treatment with 5-ASA, L-Qu, and H-Qu (Fig. 1A). Compared with the DSS model group, quercetin administration at various doses significantly alleviated body weight loss ($P < 0.05$), reduced DAI scores, increased colon length, and decreased spleen index in mice (Fig. 1B through F). The results indicate that quercetin shows similar effectiveness to 5-ASA, a clinically-proven treatment, in reducing the symptoms of colitis induced by DSS. Comparison with the NC group revealed pathological alterations in hematoxylin and eosin (H&E) staining in the DSS group, characterized by colonic mucosal damage, epithelial crypt distortion, and goblet cell detachment. H&E staining (Fig. 1G) showed that the intestinal glands in the NC group were neatly arranged without infiltration of inflammatory cells. In contrast, the DSS group exhibited irregular structures of the colonic mucosal glands with significant infiltration of inflammatory cells. High-dose quercetin treatment significantly improved the pathological changes of the colonic epithelial barrier. These findings suggested that Qu ameliorates both the clinical symptoms and histopathological features of DSS-induced colitis.

### Quercetin improved the impaired intestinal mucosal barrier and inflammation in DSS-induced mice

Tight junction proteins are key players in how UC develops. In mice with DSS-induced colitis, the amounts of occludin and claudin-1 in their colon tissue dropped noticeably (Fig. 2A). However, when we included quercetin into the mix, this effect seemed to be turned on its head, bringing those protein levels back up (Fig. 2B and C).

Inflammatory factor levels are a gage for the extent of UC. The immunohistochemical (IHC) analysis of colonic tissue, depicted in Fig. 2D through G, revealed markedly increased positive areas of the inflammatory markers IL-1β and IL-6 in the DSS group compared with the NC group. whereas quercetin treatment significantly attenuated their expression. Furthermore, we investigated the mRNA levels of inflammatory factors in the colonic tissues of mice across all groups (Fig. 2H through J). Our findings suggest that quercetin could enhance the condition of UC patients by mending the impaired intestinal lining and soothing the inflammatory turmoil.

### Quercetin alleviated DSS-induced colitis in mice in a gut microbe-dependent manner

To investigate how quercetin improves UC through the gut microbiota, we first wiped out the existing gut bacteria in mice using antibiotics a week before the experiment. The experimental design is outlined in Fig. 3A. The results showed that the Qu-FMT group experienced a notable decrease in colon shortening, weight loss, disease activity index (DAI), and spleen index compared with the DSS-FMT group (Fig. 3B through F). In addition, the Qu-FMT group also showed better pathological signs (Fig. 3G) and higher levels of occludin and claudin-1 (Fig. 3J through L). Interestingly, while the Qu group alone did not significantly lower the mRNA levels of inflammatory factors compared with the DSS-FMT group, the Qu-FMT group did show significant differences in these inflammatory marker expressions (Fig. 3H and I). These results strongly suggest that quercetin's beneficial effects on UC hinge on the presence and activity of the gut microbiota.

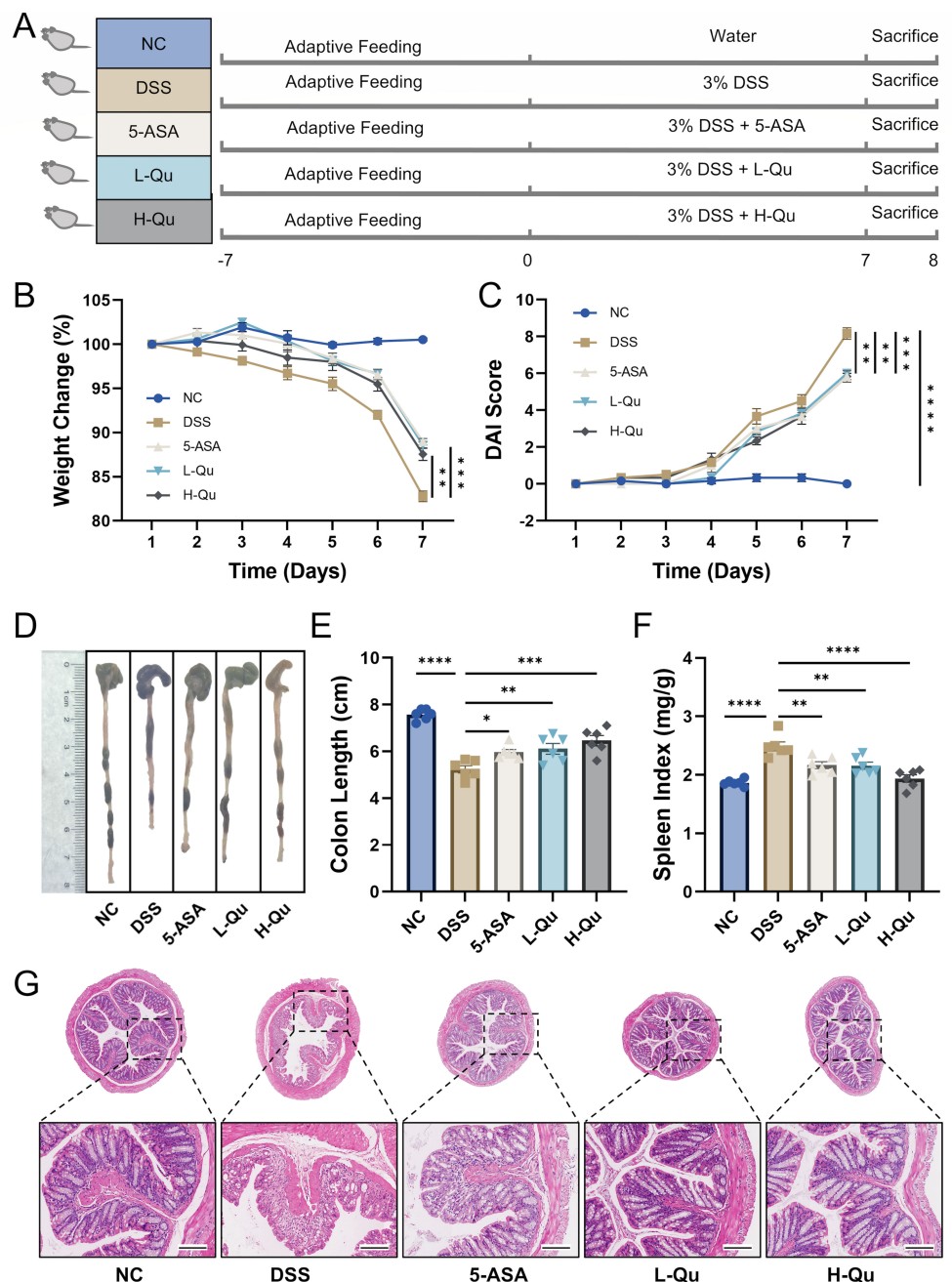

**FIG 1** Qu ameliorates DSS-induced colitis in mice. (A) Flow diagram of the experiment. (B) Body weight change (%). (C) DAI score. (D) Colon image. Histogram statistic of colon length (E) and spleen index (F). Data were shown as mean ± SEM ($N$ = 6). (G) Representative H&E staining image of colon tissue (scale bar = 200 µm) ($N$ = 3). *$P < 0.05$, **$P < 0.01$, ***$P < 0.001$, ****$P < 0.0001$ vs the DSS group. ns, not significant.

## Quercetin regulated the composition of gut microbiota

Gut microbiota imbalance is a key factor in the progression of UC (17, 18). To elucidate the compositional changes in gut microbiota across different groups, metagenomic sequencing was utilized. As shown in Fig. 4A, this study analyzed the structural composition of the gut microbiota of mice in each group at the phylum level. The principal coordinates analysis (PCoA) at the species level, as depicted in Fig. 4B, demonstrates that DSS-induced colitis induces significant alterations in gut microbiota profiles compared with the NC groups, with Qu treatment exerting a partial modulatory

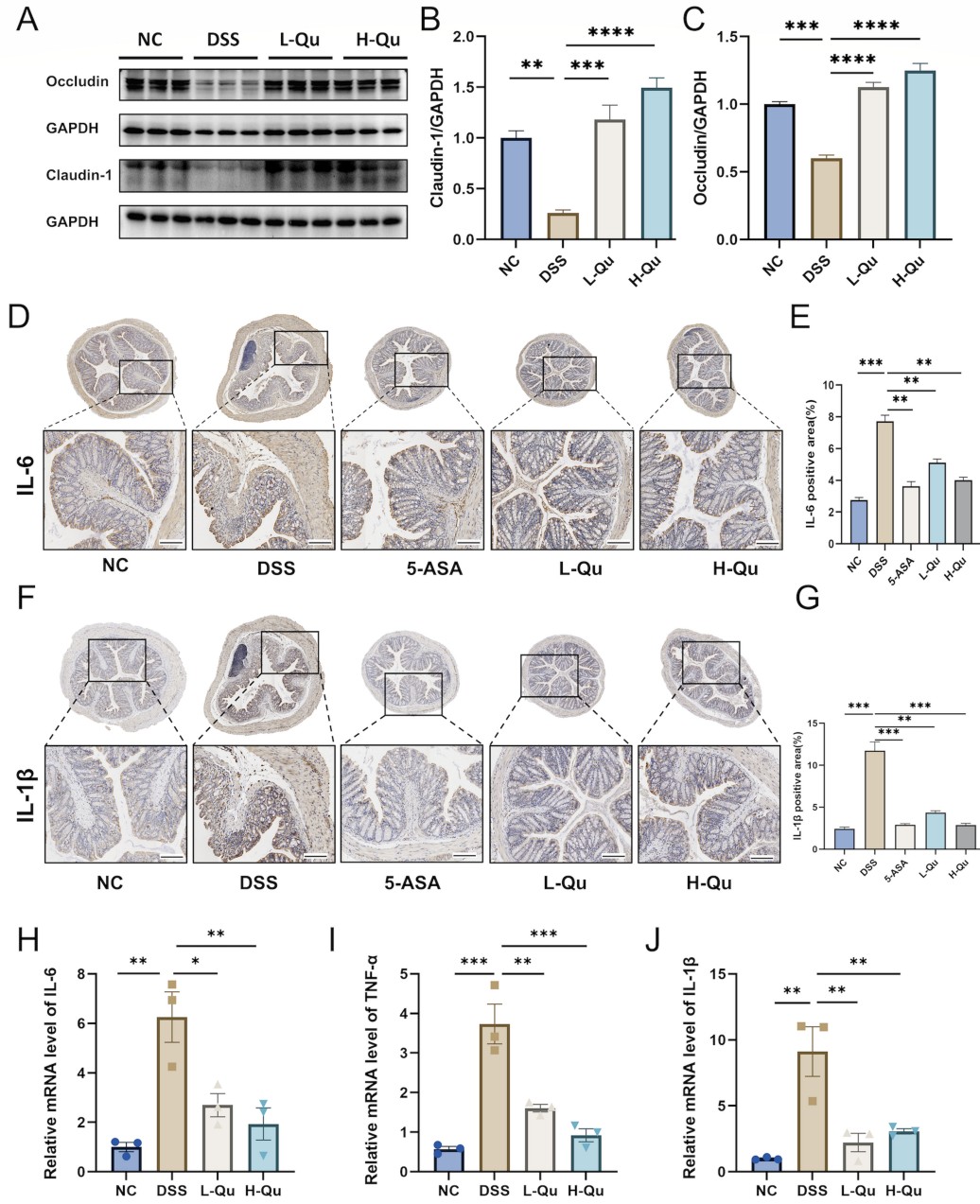

**FIG 2** Qu ameliorates intestinal barrier and inflammation in DSS-induced colitis in mice. (A–C) The protein expression of claudin-1 and occludin was detected by Western blot. Representative IHC staining images of colonic tissues showing IL-6 (D) and IL-1β (F) expression (scale bar = 200 µm). Semi-quantitative analysis of IL-6 (E) and IL-1β (G) based on IHC staining. The relative mRNA expression of IL-6 (H), TNF-α (I), and IL-1β (J) in colon tissue was detected by RT-qPCR. Data were shown as mean ± SEM ($N = 3$). *$P < 0.05$, **$P < 0.01$, ***$P < 0.001$, ****$P < 0.0001$ vs the DSS group. ns, not significant.

effect on these changes. Additionally, there was a significant decrease in the abundance of *Bacteroidota* in the NC group, whereas an increase in *Firmicutes* was observed at the phylum level in the DSS group. Consequently, this led to an elevated *Firmicutes*-to-*Bacteroidota* ratio in the DSS group, which was subsequently reversed by Qu treatment, as illustrated in Fig. 4C. LDA scores indicated that *Bacteroides*, *Alistipes*, *Mucispirillum*, and *Duncaniella* may play a significant role in DSS-induced colitis (Fig. 4D). Comparative analyses among the three groups for several specific species are illustrated in Fig. 4E through L. Collectively, these results demonstrate that quercetin administration modulates intestinal microbial composition in DSS-induced colitic mice.

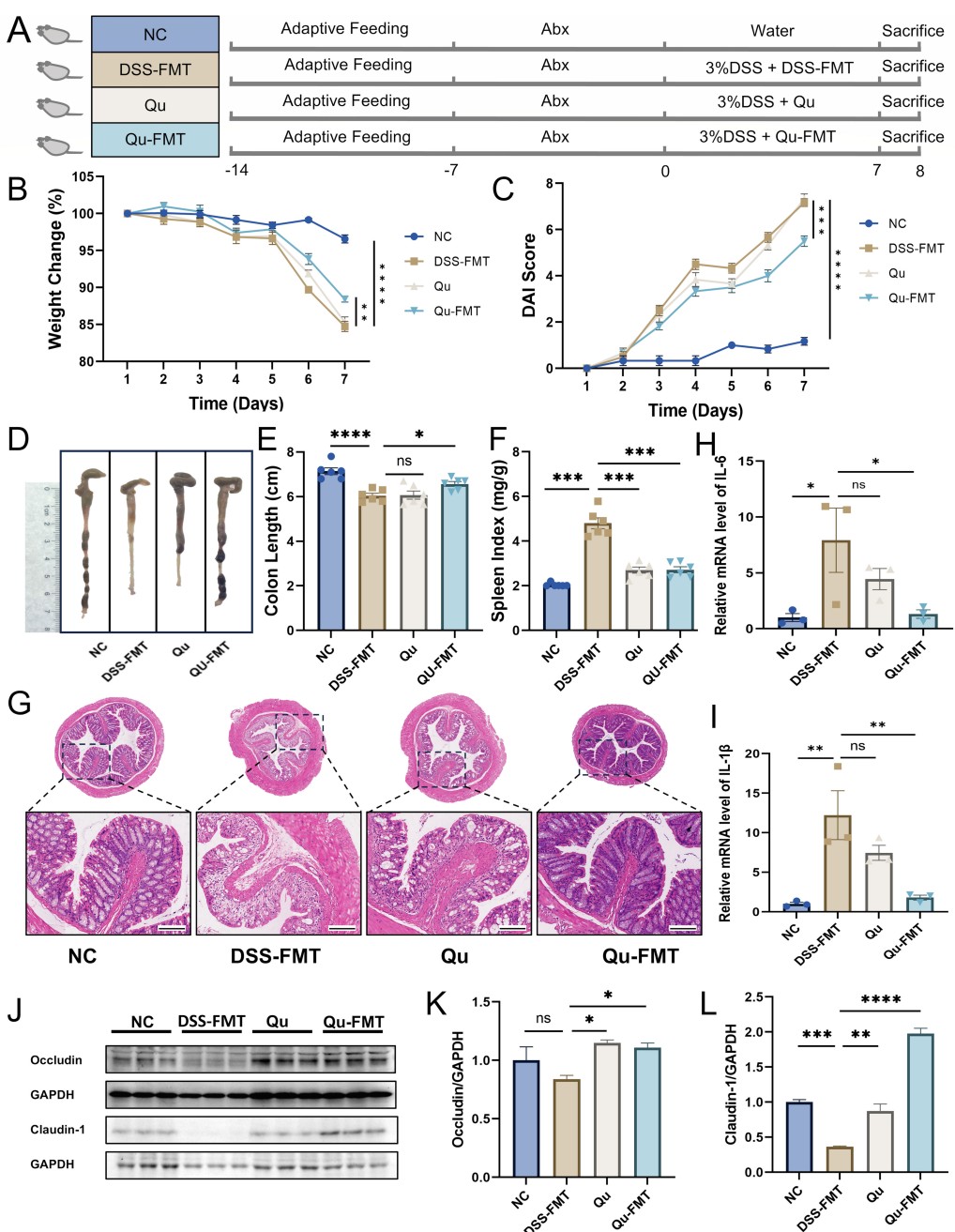

**FIG 3** Fecal microbiota transplantation from Qu-treated mice alleviated DSS-induced colitis in mice. (A) Flow diagram of the experiment. (B) Body weight change (%). (C) DAI score. (D) Colon image. Histogram statistic of colon length (E) and spleen index (F). Data were shown as mean ± SEM ($N = 6$). (G) Representative H&E staining image of colon tissue (scale bar = 200 µm). The relative mRNA expression of IL-6 (H) and IL-1β (I) in colon tissue was detected by RT-qPCR. (J–L) The protein expression of occludin and claudin-1 was detected by Western blot ($N = 3$). *$P < 0.05$, **$P < 0.01$, ***$P < 0.001$, ****$P < 0.0001$ vs the DSS group. ns, not significant.

## Quercetin altered tryptophan metabolism in DSS-induced mice

Beyond intestinal flora composition, metabolites produced by commensal microorganisms significantly contribute to IBD development (19). The tryptophan metabolic pathway is known to be significant in UC (20–22). Our study delved deeper into quercetin's potential to alleviate colitis by influencing this specific biological pathway. Using LC-MS/MS technology, we performed comprehensive metabolite profiling on

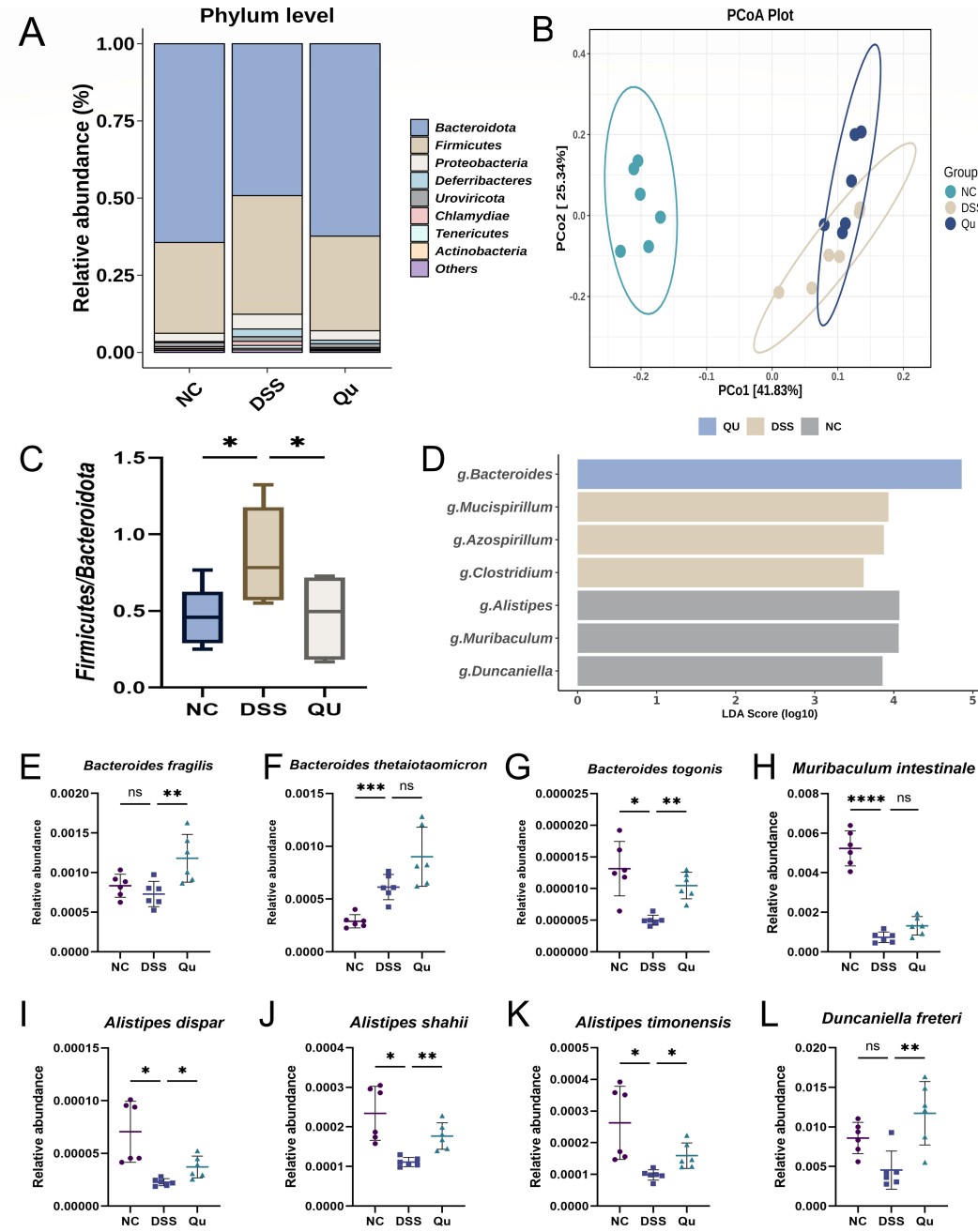

**FIG 4** Qu alleviated gut microbiota disruption in DSS-induced mice. (A) Relative abundance histograms at phylum levels in three groups. (B) PCoA analysis between three groups at species levels. (C) Firmicutes/Bacteroidota ratio. (D) LDA score with genus levels. (E–L) Difference analysis of gut microbiota at species levels. Data were shown as mean ± SEM (*N* = 6). *$P < 0.05$, **$P < 0.01$, ***$P < 0.001$, ****$P < 0.0001$ vs the DSS group. ns, not significant.

serum samples from each experimental mouse cohort. The resulting metabolic heatmap revealed striking differences in biochemical signatures between the DSS-treated control group and the quercetin intervention group (Fig. 5A). Notably, quercetin supplementation significantly elevated multiple tryptophan-derived metabolites in UC-afflicted mice, including indole and its derivatives (ILA, indoleacetic acid, indole-3-propionic acid, indoleacrylic acid) along with 5-hydroxyindoleacetic acid (Fig. 5C through J). To uncover possible mechanistic connections, we performed correlation analyses mapping microbial population dynamics against tryptophan metabolism products (Fig. 5B). This analysis

pinpointed *Bacteroides* as showing the strongest association with ILA levels, prompting us to focus subsequent investigations on this particular metabolite.

## ILA alleviated colitis in DSS-induced mice

To investigate the potential therapeutic effects of exogenous ILA supplementation in DSS-induced colitis mice, we conducted systematic intervention experiments. A second cohort of mice was subsequently established, comprising the following experimental groups: NC group, DSS group, and ILA group (Fig. 6A). The ILA group exhibited increased colon length and mouse body weight, decreased DAI and spleen index, and reduced pathological changes compared with the DSS group (Fig. 6B through G). Notably, as ILA is a natural ligand for AhR, we also examined its effect on AhR in this study. Consequently, we chose CYP1B1, a downstream target of AhR, as the observational index (Fig. 6J through L). In addition, the experimental results showed that ILA could significantly reduce the mRNA expression levels of TNF-α and IL-1β in the colonic tissues of mice in the UC group (Fig. 6H and I). The findings suggested that ILA supplementation is beneficial for colitis in DSS-induced mice.

## Quercetin activated AhR pathway to relieve colitis in DSS-induced mice

Tryptophan metabolites have been shown to regulate intestinal homeostasis via AhR (23). In a further effort to investigate whether AhR activation is involved in this process, we examined the levels of the AhR downstream product CYP1B1 (24). As shown in Fig. 7A, mice were randomly assigned to one of five groups: NC group, DSS group, AhR antagonist CH223191 (CH) group, quercetin group (Qu), and a group receiving both quercetin and the AhR antagonist CH223191 (Qu + CH). We noticed a dip in CYP1B1 expression within the colonic tissues of the DSS group mice. Adding insult to injury, this decrease was even more pronounced in the group treated with the AhR antagonist CH223191. However, quercetin not only reversed this downward trend in CYP1B1 levels but also appeared to improve several aspects of UC in the mice. Specifically, it eased the clinical signs of UC, improved the condition of colonic tissue under microscopic examination, boosted the expression of occludin and claudin-1, and tweaked the mRNA levels of inflammatory factors (Fig. 7B through L). That being said, the Qu + AhR antagonist CH223191 group did not fare quite as well as the Qu group. This suggests that quercetin's ability to restore intestinal mucosal barrier function in these DSS-induced mice hinges on the activation of AhR.

## DISCUSSION

This study investigated the therapeutic potential of quercetin in treating UC and examined its underlying mechanisms. The results showed that quercetin administration improved key disease markers in DSS-induced mice, including increased body weight, longer colon length, lower DAI scores, and enhanced colon tissue integrity. Furthermore, the research uncovered significant gut microbiota dysbiosis and a marked deficiency in the tryptophan-derived metabolite ILA in UC-afflicted mice. The study also explored quercetin's multifaceted therapeutic actions, which appear to involve restoring balanced gut microbiota, regulating tryptophan metabolism, and activating the AhR pathway.

Quercetin, a flavonoid herbal extract, is known for its significant anti-inflammatory effects (25, 26), and previous studies have indicated its potential benefit in UC (16). Consistent with earlier findings, we therefore adopted two effective doses of quercetin (50 and 100 mg/kg) in our colitis model (27, 28). However, the mechanisms have mainly focused on its intestinal immunomodulatory functions. At present, the full pathogenesis underlying UC has yet to be comprehensively clarified (29), with the restoration of microbiota disorders emerging as a potential treatment target. A reduction in the populations of *Mycobacterium anisopliae*, *Bifidobacterium bifidum*, *Lactobacillus spp.*, and *Clostridium* spp. was observed in a colitis mouse model of *Mycobacterium citrinum*. In contrast, the quercetin-treated group exhibited an increase in the populations of these microorganisms (30). Notably, our study detected a decline in the biodiversity of the

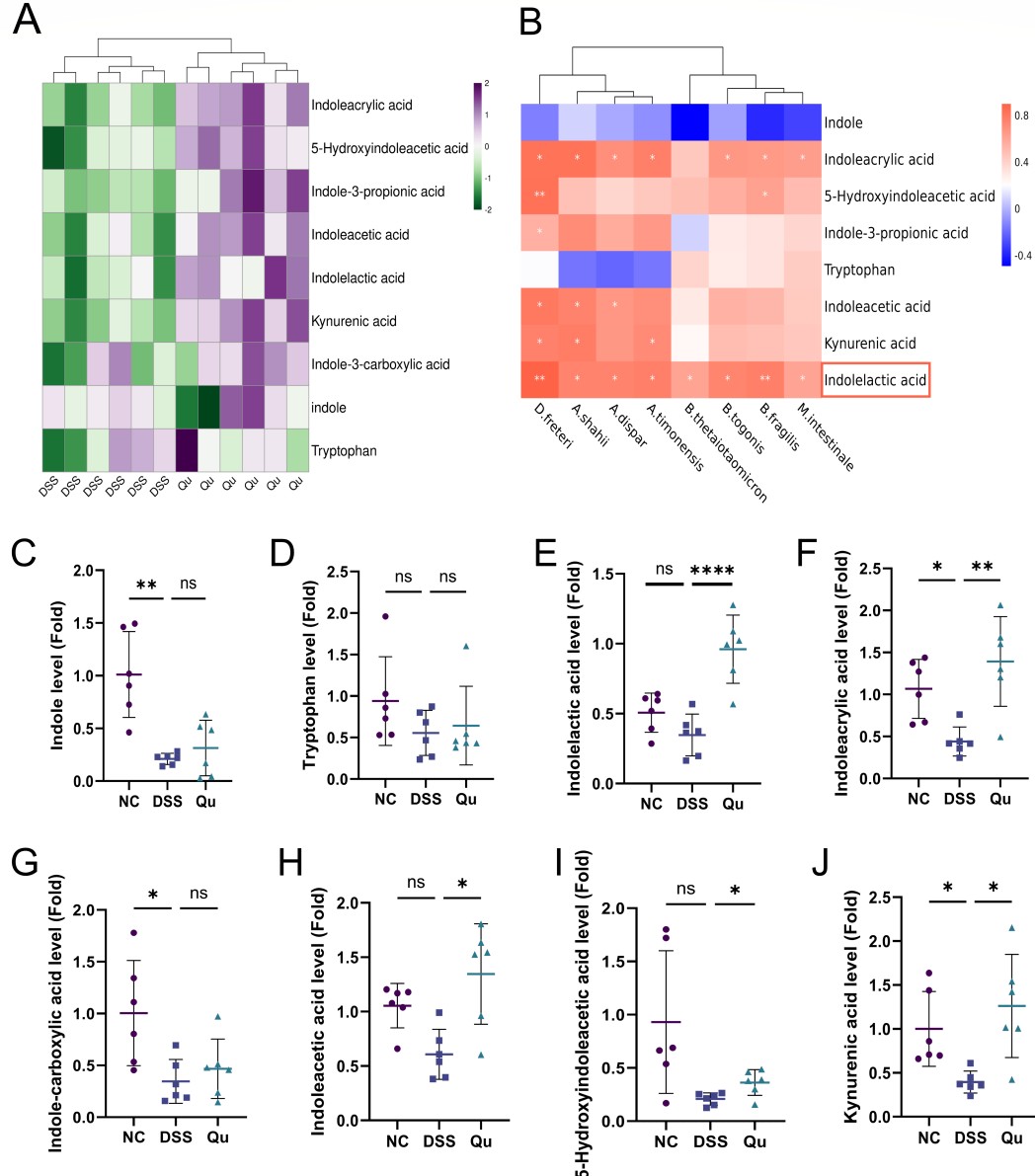

**FIG 5** Qu regulates tryptophan metabolism by targeting gut microbiota. (A) Heatmap of tryptophan-related metabolites. (B) Correlation heatmap of tryptophan-related metabolites and intestinal microbiota. (C–J) Relative abundance of tryptophan-related metabolites in three groups. Data were shown as mean ± SEM ($N = 6$). *$P < 0.05$, **$P < 0.01$, ***$P < 0.001$, ****$P < 0.0001$ vs the DSS group. ns, not significant.

microbiota in the DSS-treated mouse group. Interestingly, quercetin was shown to have the capacity to counteract this disruption in the microbiota composition. Additionally, a previous study has shown that the up-regulation of the abundance of *B. thetaiotaomicron* in the genus Bacteroidetes of the microbiota can alleviate intestinal inflammation in DSS model mice (31, 32). This adds further support to the potential therapeutic mechanisms of quercetin in UC.

Moreover, gut microbial metabolite levels are altered in individuals with UC. Numerous studies have established the therapeutic potential of tryptophan-derived metabolites, bile acids, and short-chain fatty acids in mitigating DSS-induced colitis in murine models (33–35). A previous study has shown a significant reduction in tryptophan metabolism levels in mice with experimental colitis (36). In this study, a reduction in a series of tryptophan metabolites was observed in the DSS group of mice. The

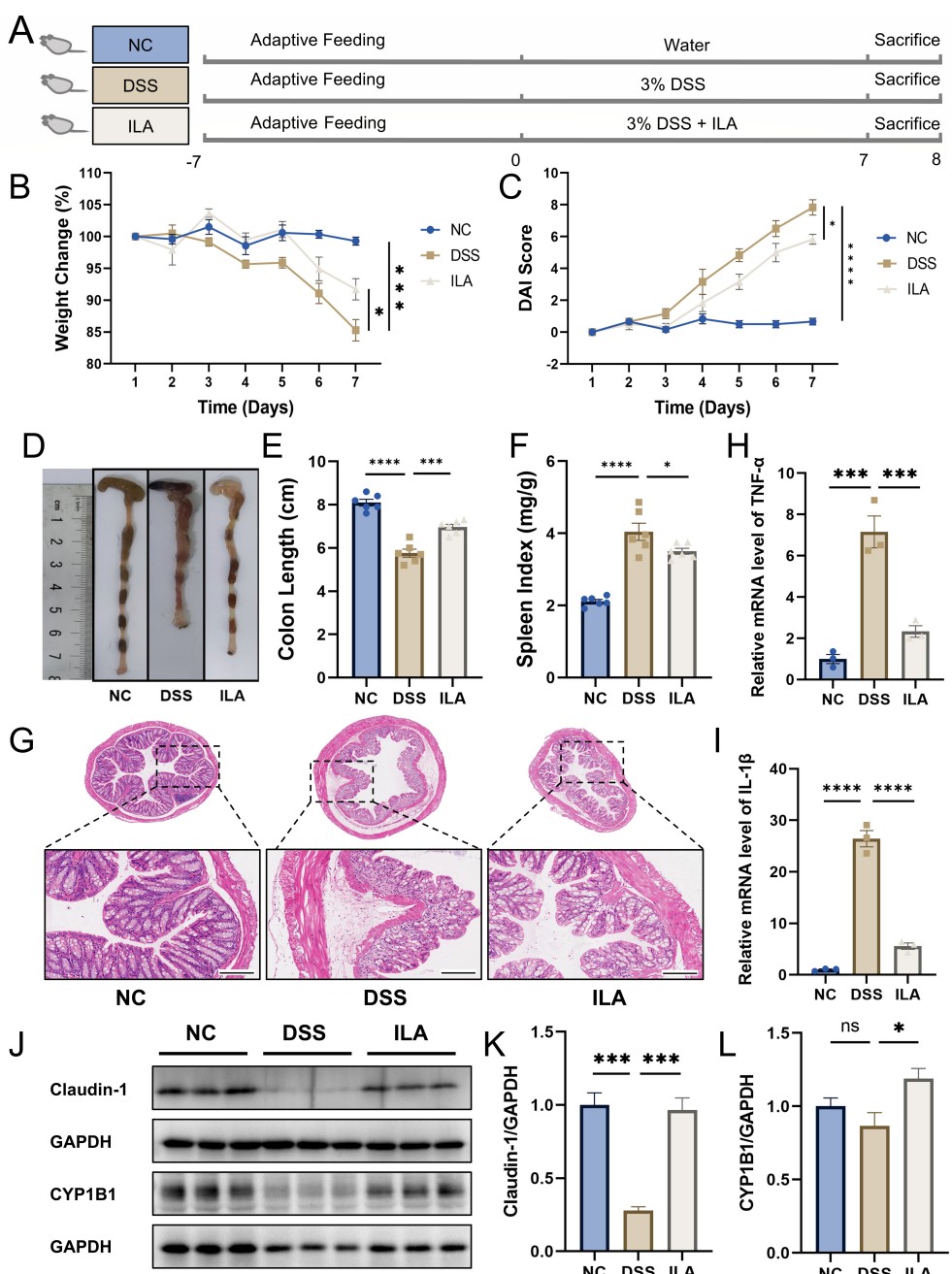

**FIG 6** ILA alleviated DSS-induced colitis in mice. (A) Flow diagram of the experiment. (B) Body weight change (%). (C) DAI score. (D) Colon image. Histogram statistic of colon length (E) and spleen index (F). Data were shown as mean ± SEM ($N = 6$). (G) Representative H&E staining image of colon tissue (scale bar = 200 μm). The relative mRNA expression of TNF-α (H) and IL-1β (I) in colon tissue was detected by RT-qPCR. (J–L) The protein expression of claudin-1 and CYP1B1 was detected by Western blot ($N = 3$). *$P < 0.05$, **$P < 0.01$, ***$P < 0.001$, ****$P < 0.0001$ vs the DSS group. ns, not significant.

administration of quercetin was found to counteract this decrease in metabolite levels. ILA showed significant changes in both pre- and post-levels across all three groups. Correlation analysis indicated a positive correlation only between ILA and Bacteriophage polymorphis. It has been demonstrated that ILA supplementation can alleviate colitis in experimental mice (9). Our observations revealed that additional ILA supplementation reversed the severity of clinical symptoms and colonic pathologic damage induced by DSS in mice. Furthermore, the results showed an increased expression of tight junction

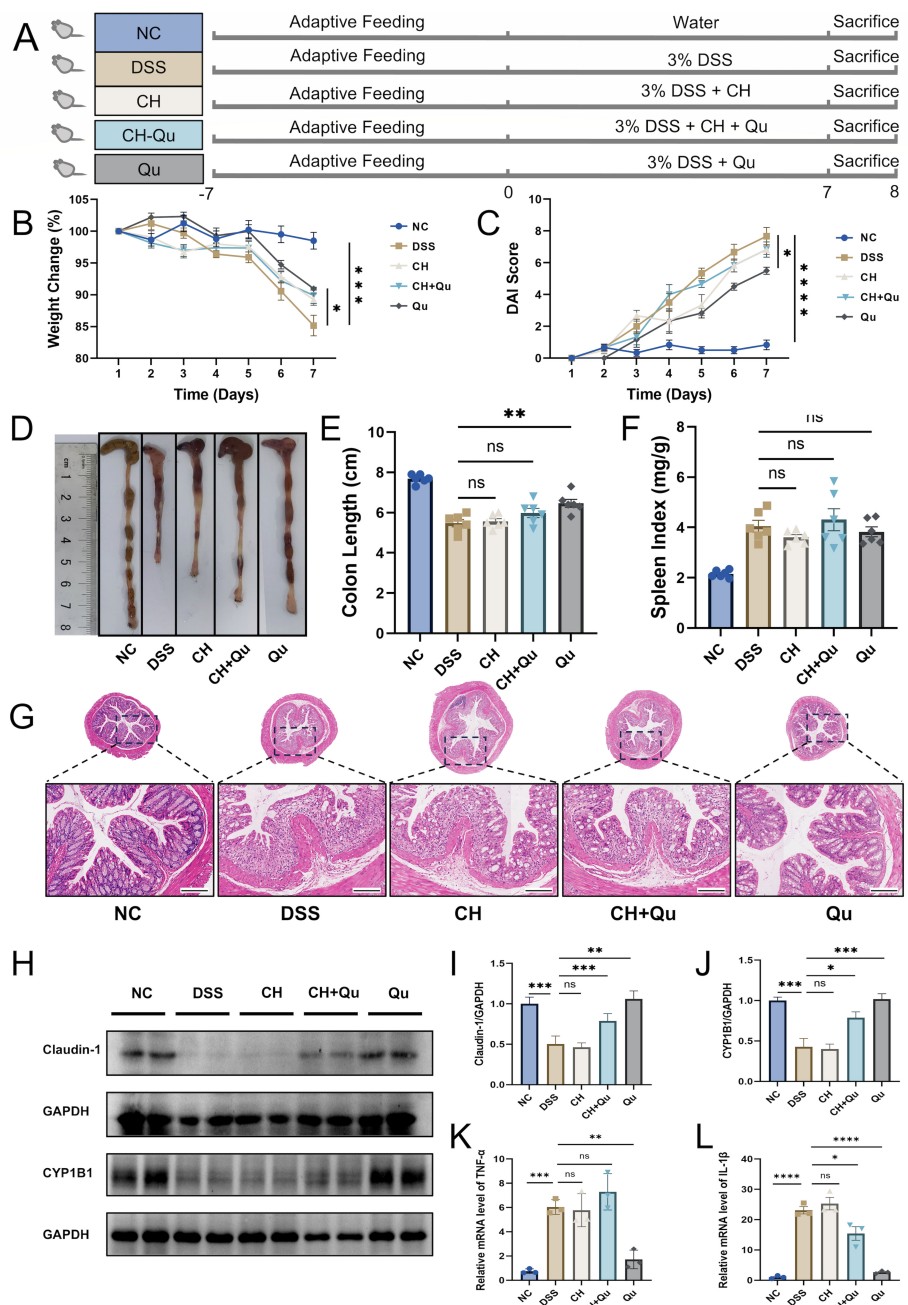

**FIG 7** Qu through ILA-AhR alleviated DSS-induced colitis in mice. (A) Flow diagram of the experiment. (B) Body weight change (%). (C) DAI score. (D) Colon image. Histogram statistic of colon length (E) and spleen index (F). Data were shown as mean ± SEM ($N = 6$). (G) Representative H&E staining image of colon tissue (scale bar = 200 µm). (H–J) The protein expression of claudin-1 and CYP1B1 was detected by western blot ($N = 3$). The relative mRNA expression of TNF-α (K) and IL-1β (L) in colon tissue was detected by RT-qPCR ($N = 3$). *$P < 0.05$, **$P < 0.01$, ***$P < 0.001$, ****$P < 0.0001$ vs the DSS group. ns, not significant.

proteins, indicating that ILA can restore mucosal barrier function and ameliorate colitis in mice.

The role of AhR in regulating intestinal barrier dysfunction in UC has received increasing attention in recent times (37). In the intestinal epithelial cells of UC patients, a decrease in AhR expression has been noted. Furthermore, in the gut's internal habitat, the activation of AhR plays a pivotal role in sustaining the integrity of the intestinal mucosal protective shield (38, 39). Consistent with previous studies (40–42), we observed

a reduction in CYP1B1 and a downstream product of AhR in the DSS group. In contrast to the DSS-treated mice, those administered the AhR antagonist exhibited reduced body weight, shorter colonic length, and more severe pathological damage in the colon. Furthermore, there was a marked decrease in the expression of the tight junction proteins Occludin and Claudin-1. These results indicate that blocking AhR signaling could exacerbate UC. When comparing treatment outcomes, the Qu + AhR antagonist group demonstrated poorer clinical efficacy and histopathological findings than the quercetin-only group. This suggests that quercetin's protective effects against colitis may depend, at least partially, on its ability to activate the AhR pathway.

To summarize, our research revealed that quercetin successfully repaired the compromised intestinal barrier in mice with DSS-induced colitis. This therapeutic effect was achieved by modulating gut bacteria to influence tryptophan metabolism, specifically increasing levels of ILA. The resulting activation of the AhR pathway helped alleviate UC symptoms. These findings provide valuable insights into how quercetin exerts its protective effects against UC, expanding our knowledge of its potential clinical applications.

## ACKNOWLEDGMENTS

This study was supported by Guangdong Medical Science and Technology Research Foundation, China (202211921445862), the Scientific Research Foundation of the Higher Education Institutions of Guangdong Province (2023KQNCX016), Guangdong Provincial Administration of Traditional Chinese Medicine Scientific Research Project (20261093), University - Hospital Joint Fund Project of Guangzhou University of Chinese Medicine (GZYSE2024Y02), Hospital Chinese Medicine Development Projects for Developing Strong Traditional Chinese Medicine Province of Guangdong Provincial Administration of TCM, China, Military-Civilian Integration and Science and Technology Innovation Bureau of Foshan (2018AG100091).

M.X. was responsible for the experimental design. F.Z. and Z.L. drafted the manuscript. M.X., W.K., and Z.Z. contributed to the experiments. F.Z., R.T., X.D., N.W., X.W., M.F., and Y.L. performed the data analysis and drew figures. All authors contributed to the article.

## AUTHOR AFFILIATIONS

[1]Guangzhou University of Chinese Medicine, Guangzhou, PR China
[2]The Seventh Affiliated Hospital, Sun Yat-sen University, Shenzhen, PR China
[3]Department of Laboratory Medicine, Guangdong Provincial Second Hospital of Traditional Chinese Medicine (Guangdong Provincial Engineering Technology Research Institute of Traditional Chinese Medicine), Guangzhou, PR China
[4]School of Basic Medical Sciences, Guangzhou Medical University, Guangzhou, PR China
[5]Department of Pharmacy, Affiliated Hospital of Qingdao University, Qingdao, Shandong, PR China
[6]The Eighth School of Clinical Medicine, Guangzhou University of Chinese Medicine, Foshan, PR China

## AUTHOR ORCIDs

Wenli Kuang  http://orcid.org/0009-0001-3967-6353
Bin Zhang  http://orcid.org/0009-0001-2493-089X
Zhaofu Zhang  http://orcid.org/0009-0007-8169-767X
Fanghao Zheng  http://orcid.org/0000-0002-7073-7847

## DATA AVAILABILITY

The metagenomic sequencing data have been deposited in the National Microbiology Data Center (NMDC, https://nmdc.cn) under accession numbers NMDC40093190-

NMDC40093207. Other supporting data are available within the article or from the corresponding authors upon reasonable request.

## ADDITIONAL FILES

The following material is available online.

### Open Peer Review

**PEER REVIEW HISTORY (review-history.pdf).** An accounting of the reviewer comments and feedback.

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
