## [Reviewer comments · mSystems]

Quercetin Alleviates ulcerative colitis via regulating gut microbiota and tryptophan metabolism

Man Xiong, Wenli Kuang, Zhenyang Liu, Ruoyi Tong, Xu Deng, Nan Wang, Xingzhi Wan, Mengyuan Feng, Yan Luo, Bin Zhang, Zhaofu Zhang, and Fanghao Zheng

Corresponding Author(s): Fanghao Zheng, Foshan Hospital of Traditional Chinese Medicine

Review Timeline:

Submission Date:	June 10, 2025
Editorial Decision:	August 4, 2025
Revision Received:	September 10, 2025
Accepted:	October 8, 2025

Editor: Hiutung Chu

Reviewer(s): Disclosure of reviewer identity is with reference to reviewer comments included in decision letter(s). The following individuals involved in review of your submission have agreed to reveal their identity: Zibin Lu (Reviewer #1); Qin Hai Ma (Reviewer #2)

Transaction Report:

DOI: <https://doi.org/10.1128/msystems.00703-25>

Re: mSystems00703-25 (**Quercetin Alleviates ulcerative colitis via regulating gut microbiota and tryptophan metabolism**)

Dear Mr. Fanghao Zheng:

Revision Guidelines

Sincerely,
Hiutung Chu
Editor
mSystems

Reviewer #1 (Comments for the Author):

This study systematically investigates the mechanism by which quercetin alleviates DSS-induced colitis in mice through the "gut microbiota-tryptophan metabolism-AhR pathway" axis. The experimental design is rigorous, the data are comprehensive, and the research holds significant scientific importance and potential clinical translational value. The study innovatively reveals the molecular mechanism by which quercetin enhances intestinal barrier function through increasing indole-3-lactic acid (ILA) levels to activate the AhR pathway. However, the following issues remain to be addressed.

1. Figure 2 presents immunohistochemical staining of IL-1 β and IL-6 in colonic tissues, but the differences in tissue expression

are not visually intuitive due to the lack of semi-quantitative analysis. We recommend that the authors supplement these data to strengthen the findings.

2. The manuscript states of the Statistical Analysis section that data are presented as mean {plus minus} standard deviation (SD), while the figure legends consistently report mean {plus minus} standard error of the mean (SEM). This discrepancy in statistical reporting metrics requires careful attention and clarification.

3. The serum metabolomics analysis in Figure 5 presents comparative data between DSS and quercetin-treated groups for tryptophan metabolites, but not with the normal control (NC) group. The authors are expected to provide a well-justified explanation for this omission.

4. In the writing section, abbreviations that have been previously defined in the text can be used directly without restating the full term.

Reviewer #2 (Comments for the Author):

This manuscript explores the protective effects of quercetin on a DSS-induced colitis model, focusing on the gut microbiota and tryptophan metabolism. This is an interesting study, and focuses on microbiome-metabolite-host interactions in inflammatory bowel disease. While the manuscript is promising, several points require clarification:

1. There is some redundancy in the introduction. The sentence "Key tryptophan metabolites-including indole, indolelactic acid (ILA), and indole-3-carbinol-function as natural activators of the aromatic hydrocarbon receptor (AhR)" is repeated with slightly different wording in the next sentence. Please combine or rephrase these sentences to improve conciseness and readability.

2. The manuscript includes two dosing levels of quercetin (L-Qu and H-Qu). Why was a conventional three-dose design not used? Please clarify how the specific doses were determined.

3. In Fig.1, both low and high doses of quercetin were evaluated. However, for the metagenomic (Fig.4) and metabolomic (Fig.5) analyses, only a single quercetin group (Qu) was used. Please state which dose was chosen and why data were collected at only one dose.

Reply to reviewers' comments

Quercetin Alleviates ulcerative colitis via regulating gut microbiota and tryptophan metabolism

Dear Prof. Hiutung Chu,

Re: mSystems00703-25

Thank you for giving us the opportunity to submit a revised version of our manuscript to *mSystems*. We sincerely appreciate the valuable time and effort you and the reviewers' have devoted to evaluating our work.

We are grateful to the reviewers for the insightful comments and constructive suggestions. These have been extremely helpful in improving the quality, clarity, and overall presentation of the manuscript. We have carefully addressed each point raised, and corresponding revisions have been made in the manuscript. All changes are highlighted in the revised version for each of review.

Below, we provide a point-by-point response to each reviewer's comments. We believe that these changes have significantly strengthened the manuscript.

Yours sincerely,

Fanghao Zheng

Reviewer #1 (Comments for the Author):

This study systematically investigates the mechanism by which quercetin alleviates DSS-induced colitis in mice through the "gut microbiota-tryptophan metabolism-AhR pathway" axis. The experimental design is rigorous, the data are comprehensive, and the research holds significant scientific importance and potential clinical translational value. The study innovatively reveals the molecular mechanism by which quercetin enhances intestinal barrier function through increasing indole-3-lactic acid (ILA) levels to activate the AhR pathway. However, the following issues remain to be addressed.

1. Figure 2 presents immunohistochemical staining of IL-1 β and IL-6 in colonic tissues, but the differences in tissue expression are not visually intuitive due to the lack of semi-quantitative analysis. We recommend that the authors supplement these data to strengthen the findings.

Answer: Thank you for the reviewers' suggestions. We have now supplemented the semi-quantitative analysis of IL-1 β and IL-6 immunohistochemical staining as requested, and the results are shown in Figures 2E and 2G.

Figures 2 and Figure Legend section:

Figure 2. Representative IHC staining images of colonic tissues showing IL-6 (D) and IL-1 β (F) expression (scale bar = 200 μ m). Semi-quantitative analysis of IL-6 (E) and IL-1 β (G) based on IHC staining.

Page 14 (Result section):

Inflammatory factor levels are a gauge for the extent of UC. The immunohisto-chemical (IHC) analysis of colonic tissue, depicted in Figures 2D-G, revealed markedly increased positive areas of the inflammatory markers IL-1 β and IL-6 in the DSS group compared with the NC group. whereas quercetin treatment significantly attenuated their expression.

2. The manuscript states of the Statistical Analysis section that data are presented as mean standard deviation (SD), while the figure legends consistently report mean standard error of the mean (SEM). This discrepancy in statistical reporting metrics requires careful attention and clarification.

Answer: Thank you for the reviewers' feedback. In our manuscript, the data were presented as mean \pm standard error of the mean (SEM). Due to an oversight, the Statistical Methods section incorrectly stated that the results were expressed as mean \pm standard deviation (SD). This error has now been corrected.

Page 12 (Statistical Methods section):

we've reported our findings as the mean \pm standard error of the mean (mean \pm SEM).

3. The serum metabolomics analysis in Figure 5 presents comparative data between DSS and quercetin-treated groups for tryptophan metabolites, but not with the normal control (NC) group. The authors are expected to provide a well-justified explanation for this omission.

Answer: We thank the reviewer for their comment. Since the primary objective of this study was to examine the changes in tryptophan metabolites in DSS-induced mice before and after quercetin treatment, the serum metabolomics data presented here focused specifically on comparisons between these two groups.

4. In the writing section, abbreviations that have been previously defined in the text can be used directly without restating the full term.

Answer: Thank you for the reviewers' suggestion. We have carefully reviewed the abbreviations throughout the manuscript to ensure that full terms are not unnecessarily repeated. For example, the term "ulcerative colitis" was removed and only the abbreviation "UC" was retained.

Page 14 (Result section):

Our findings suggest that quercetin could enhance the condition of UC patients by mending the impaired intestinal lining and soothing the inflammatory turmoil.

To investigate how quercetin improves UC through the gut microbiota.

Page 21 (Discussion section):

These results indicate that blocking AhR signaling could exacerbate UC.

The resulting activation of the AhR pathway helped alleviate UC symptoms.

Reviewer #2 (Comments for the Author):

This manuscript explores the protective effects of quercetin on a DSS-induced colitis model, focusing on the gut microbiota and tryptophan metabolism. This is an interesting study, and focuses on microbiome-metabolite-host interactions in inflammatory bowel disease. While the manuscript is promising, several points require clarification:

1. There is some redundancy in the introduction. The sentence "Key tryptophan metabolites-including indole, indolelactic acid (ILA), and indole-3-carbinol-function as natural activators of the aromatic hydrocarbon receptor (AhR)" is repeated with slightly different wording in the next sentence. Please combine or rephrase these sentences to improve conciseness and readability.

Answer: We thank the reviewer for highlighting the redundancy in the introduction. We agree that the two sentences contained overlapping information about how tryptophan metabolites activate the AhR. To improve clarity and concision, we have combined and rephrased these sentences as follows:

Page 5 (Introduction section):

Beyond the bacterial populations themselves, microbial products also play a significant role in the development of inflammatory bowel disease (IBD). Tryptophan-derived metabolites, such as indole, indolelactic acid (ILA), and indole-3-carbinol, act as endogenous ligands of the aryl hydrocarbon receptor (AhR). Binding to the AhR activates the receptor, which is expressed in intestinal epithelial cells and recognized as essential for maintaining the integrity of the intestinal mucosal barrier (9, 10).

2.The manuscript includes two dosing levels of quercetin (L-Qu and H-Qu). Why was a conventional three-dose design not used? Please clarify how the specific doses were determined.

Answer: We appreciate this valuable comment. A conventional three-dose design is indeed common in pharmacological studies; however, in our study we selected two representative doses (50 mg/kg and 100 mg/kg) for the following reasons.

First, previous studies in DSS-induced colitis models tested quercetin at 25, 50,100 mg/kg, and demonstrated that both 50 mg/kg and 100 mg/kg significantly improved disease activity index (DAI) index, colon length, and histopathological outcomes, while a lower dose (25 mg/kg) showed limited therapeutic efficacy (27, 28). Based on these findings, we considered 50 mg/kg as effective low dose and 100 mg/kg as a higher dose with consistent benefit.

Second, our intention was to minimize unnecessary animal use, in accordance with the 3R principles of animal welfare. For those reasons, we believe that a two-dose design was sufficient to address our study objective.

Page 19 (Discussion section):

Quercetin, a flavonoid herbal extract, is known for its significant anti-inflammatory effects (25, 26), and previous studies have indicated its potential benefit in UC (16). Consistent with earlier findings, we therefore adopted two effective doses of quercetin (50 mg/kg and 100 mg/kg) in our colitis model (27, 28). However, the mechanisms have mainly focused on its intestinal immunomodulatory functions.

3. In Fig.1, both low and high doses of quercetin were evaluated. However, for the metagenomic (Fig.4) and metabolomic (Fig.5) analyses, only a single quercetin group (Qu) was used. Please state which dose was chosen and why data were collected at only one dose.

Answer: We thank the reviewer for this valuable comment. To further explore the mechanistic effects of quercetin, we focused the metagenomic (Fig.4) and metabolomic (Fig.5) analyses on high-dose quercetin group (100 mg/kg). although the low-dose (50 mg/kg) and high-dose groups exhibited similar improvements in weight change, DAI index, and colon length (shown in Figures 1B-F), the high-dose group showed higher expression of intestinal barrier proteins (shown in Figures 2A-C), indicating a stronger therapeutic effect.

In addition, due to limitations in sample availability, and potential variability in metabolite measurements from stored specimens, as well as the high cost and complexity of multi-omics analyses, we focused on the high-dose group to ensure robust and reliable mechanistic data. Based on that, we capture the key microbiota and metabolite changes associated with quercetin treatment in UC.

Reference:

27. Wang X, Xie X, Li Y, Xie X, Huang S, Pan S, et al. Quercetin Ameliorates Ulcerative Colitis by Activating Aryl Hydrocarbon Receptor to Improve Intestinal Barrier Integrity. *Phytotherapy research : PTR* (2024) 38(1):253-64. Epub 2023/10/24. doi: 10.1002/ptr.8027.
28. Salaritabar A, Darvishi B, Hadjiakhoondi F, Manayi A, Sureda A, Nabavi SF, et al. Therapeutic Potential of Flavonoids in Inflammatory Bowel Disease: A Comprehensive Review. *World journal of gastroenterology* (2017) 23(28):5097-114. Epub 2017/08/16. doi: 10.3748/wjg.v23.i28.5097.

In total, we sincerely appreciate for you and the reviewers their careful examinations and deep thoughts. These precious comments have significantly enhanced the quality of the manuscript.

Re: mSystems00703-25R1 (**Quercetin Alleviates ulcerative colitis via regulating gut microbiota and tryptophan metabolism**)

Dear Mr. Fanghao Zheng:

Your manuscript has been accepted, and I am forwarding it to the ASM production staff for publication. Your paper will first be checked to make sure all elements meet the technical requirements. ASM staff will contact you if anything needs to be revised before copyediting and production can begin. Otherwise, you will be notified when your proofs are ready to be viewed.

Sincerely,
Hiutung Chu
Editor
mSystems

Reviewer #1 (Comments for the Author):

The author's revision answered my question.

Reviewer #2 (Comments for the Author):

It has been made the necessary revisions based on the suggestions and it can be accepted.